# Transformation of Node to Knowledge Graph Embeddings for Faster Link Prediction in Social Networks

Archit Parnami[*,1], Mayuri Deshpande[2], Anant Kumar Mishra[2], and Minwoo Lee[1]

[1] The University of North Carolina at Charlotte, NC, USA
aparnami@uncc.edu, minwoo.lee@uncc.edu
[2] Siemens Corporate Technology, Charlotte, NC, USA

**Abstract.** Recent advances in neural networks have solved common graph problems such as link prediction, node classification, node clustering, node recommendation by developing embeddings of entities and relations into vector spaces. Graph embeddings encode the structural information present in a graph. The encoded embeddings then can be used to predict the missing links in a graph. However, obtaining the optimal embeddings for a graph can be a computationally challenging task specially in an embedded system. Two techniques which we focus on in this work are 1) node embeddings from random walk based methods and 2) knowledge graph embeddings. Random walk based embeddings are computationally inexpensive to obtain but are sub-optimal whereas knowledge graph embeddings perform better but are computationally expensive. In this work, we investigate a transformation model which converts node embeddings obtained from random walk based methods to embeddings obtained from knowledge graph methods directly without an increase in the computational cost. Extensive experimentation shows that the proposed transformation model can be used for solving link prediction in real-time.

**Keywords:** Knowledge Graphs · Node Embeddings · Link Prediction.

## 1 INTRODUCTION

With the advancement in internet technology, online social networks have become part of people's everyday life. Their analysis can be used for targeted advertising, crime detection, detection of epidemics, behavioural analysis etc. Consequently, a lot of research has been devoted to computational analysis of these networks as they represent interactions between a group of people or community and it is of great interest to understand these underlying interactions. Generally, these networks are modeled as graphs where a node represents people or entity and an edge represent interactions, relationships or communication

---

[*] Work done while A. Parnami was as an intern at Siemens.

between two of them. For example, in a social network such as Facebook and Twitter, people are represented by nodes and the existence of an edge between two nodes would represent their friendship. Other examples would include a network of products purchased together on an E-commerce website like Amazon, a network of scientists publishing in a conference where an edge would represent their collaboration or a network of employees in a company working on a common project.

Inherent nature of social networks is that they are dynamic, i.e., over time new edges are added as a network grows. Therefore, understanding the likelihood of future association between two nodes is a fundamental problem and is commonly known as *link prediction* [19]. Concretely, link prediction is to predict whether there will be a connection between two nodes in the future based on the existing structure of the graph and the existing attribute information of the nodes. For example, in social networks, link prediction can suggest new friends; in E-commerce, link prediction can recommend products to be purchased together [11]; in bioinformatics, it can find interaction between proteins [2]; in co-authorship networks, it can suggest new collaborations and in the security domain, link prediction can assist in identifying hidden groups of terrorists or criminals [3].

Over the years, a large number of link prediction methods have been proposed [21]. These methods are classified based on different aspects such as the network evolution rules that they model, the type and amount of information they used or their computational complexity. Similarity-based methods such as Common Neighbors [19], Jaccard's Coefficient, Adamic-Adar Index [1], Preferential Attachment [4], Katz Index [16] use different graph similarity metrics to predict links in a graph. Embedding learning methods [18,2,13,25] take a matrix representation of the network and factorize them to learn a low-dimensional latent representation/embedding for each node. Recently proposed network embeddings such as DeepWalk [25] and node2vec [13] are in this category since they implicitly factorize some matrices [27].

Similar to these node embedding methods, recent years have also witnessed a rapid growth in knowledge graph embedding methods. A knowledge graph (KG) is a graph with entities of different types of nodes and various relations among them as edges. Link prediction in such a graph is known as knowledge graph completion. It is similar to link prediction in social network analysis, but more challenging because of the presence of multiple types of nodes and edges. For knowledge graph completion, we not only determine whether there is a link between two entities or not, but also predict the specific type of the link. For this reason, the traditional approaches of link prediction are not capable of knowledge graph completion. Therefore, to tackle this issue, a new research direction known as knowledge graph embedding has been proposed [24,8,31,20,15,7,28]. The main idea is to embed components of a KG including entities and relations into continuous vector spaces, so as to simplify the manipulation while preserving the inherent structure of the KG.

Neither of these two approaches, however, can generate "optimal" embeddings "quickly" for real-time link prediction on new graphs. Random walk based node embedding methods are computationally efficient but give poor results whereas KG-based methods produce optimal results but are computationally expensive. Thus, in this work, we mainly focus on embedding learning methods (i.e., Walk based node embedding methods and knowledge graph completion methods) which are capable of finding optimal embeddings quickly enough to meet real-time constraints for practical applications. To bridge the gap between computational time and performance of embeddings on link prediction, we propose the following contributions in this work:

- We compare the embedding's performance and computational cost of both Random walk based node embedding and KG-based embedding methods and empirically determine that Random walk based node embedding methods are faster but give sub-optimal results on link prediction whereas KG based embedding methods are computationally expensive but perform better on link prediction.
- We propose a transformation model that takes node embeddings from Random walk based node embedding methods and output near optimal embeddings without an increase in computational cost.
- We demonstrate the results of transformation through extensive experimentation on various social network datasets of different graph sizes and different combinations of node embeddings and KG embedding methods.

## 2   Background

### 2.1   Problem Definition

Let $G_{homo} = \langle V, E, A \rangle$ be an unweighted, undirected homogeneous graph where $V$ is the set of vertices, $E$ is the set of observed links, i.e., $E \subset V \times V$ and $A$ is the adjacency matrix respectively. The graph $G$ represents the topological structure of the social network in which an edge $e = \langle u, v \rangle \in E$ represents an interaction that took place between $u$ and $v$. Let $U$ denote the universal set containing all $(|V| \times (|V| - 1))/2$ possible edges. Then, the set of non-existent links is $U - E$. Our assumption is that there are some missing links (edges that will appear in future) in the set $U - E$. Then the link prediction task is *given the current network $G_{homo}$, find out these missing edges.*

Similarly, let $G_{kg} = \langle V, E, A \rangle$ be a Knowledge Graph (KG). A KG is a directed graph whose nodes are entities and edges are *subject-property-object* triple facts. Each edge of the form *(head entity, relation, tail entity)* (denoted as $\langle h, r, t \rangle$) indicates a relationship $r$ from entity $h$ to entity $t$. For example, $\langle Bob, isFriendOf, Sam \rangle$ and $\langle Bob, livesIn, NewYork \rangle$. Note that the entities and relations in a KG are usually of different types. Link prediction in KGs aims to predict the missing h or t for a relation fact triple $\langle h, r, t \rangle$, used in [9,6,8]. In this task, for each position of missing entity, the system is asked to rank a set

of candidate entities from the knowledge graph, instead of only giving one best result [9,8].

We then formulate the problem of link prediction on graph $G$ such that $G \equiv G_{homo} \equiv G_{kg}$, i.e., KG with only one type of entity and relation. Link prediction is then to predict the missing $h$ or $t$ for a relation fact triple $\langle h, r, t \rangle$ where both $h$ and $t$ are of same kind. For example $\langle Bob, isFriendOf, ? \rangle$ or $\langle Sam, isFriendOf, ? \rangle$.

### 2.2   Graph Embedding Methods

Graph embedding aims to represent a graph in a low dimensional space which preserves as much graph property information as possible. The differences between different graph embedding algorithms lie in how they define the graph property to be preserved. Different algorithms have different insights of the node (/edge/substructure/whole-graph) similarities and how to preserve them in the embedded space. Formally, given a graph $G = \langle V, E, A \rangle$, a node embedding is a mapping $f_1 \colon v_i \to \mathbf{y_i} \in \mathbb{R}^d \quad \forall i \in [n]$ where $d$ is the dimension of the embeddings, $n$ the number of vertices and the function $f$ preserves some proximity measure defined on graph $G$. If there are multiple types of links/relations in the graph then similar to node embeddings, relation embeddings can be obtained as $f \colon r_j \to \mathbf{y_j} \in \mathbb{R}^d \quad \forall j \in [k]$ where $k$ the number of types of relations.

**Node Embeddings using Random Walk** Random walks have been used to approximate many properties in the graph including node centrality [23] and similarity [26]. Their key innovation is optimizing the node embeddings so that nodes have similar embeddings if they tend to co-occur on short random walks over the graph. Thus, instead of using a deterministic measure of graph proximity [5], these random walk methods employ a flexible, stochastic measure of graph proximity, which has led to superior performance in a number of settings [12]. Two well known examples of random walk based methods are node2vec [13] and DeepWalk [25].

**KG Embeddings** KG embedding methods usually consists of three steps. The first step specifies the form in which entities and relations are represented in a continuous vector space. Entities are usually represented as vectors, i.e. deterministic points in the vector space [24,8,31]. In the second step, a scoring function $f_r(h, t)$ is defined on each fact $\langle h, r, t \rangle$ to measure its plausibility. Facts observed in the KG tend to have higher scores than those that have not been observed. Finally, to learn those entity and relation representations (i.e., embeddings), the third step solves an optimization problem that maximizes the total plausibility of observed facts as detailed in [30]. KG embedding methods which we use for experiments in this paper are TransE [8], TransH [31], TransD [20], RESCAL [32] and SimplE [17].

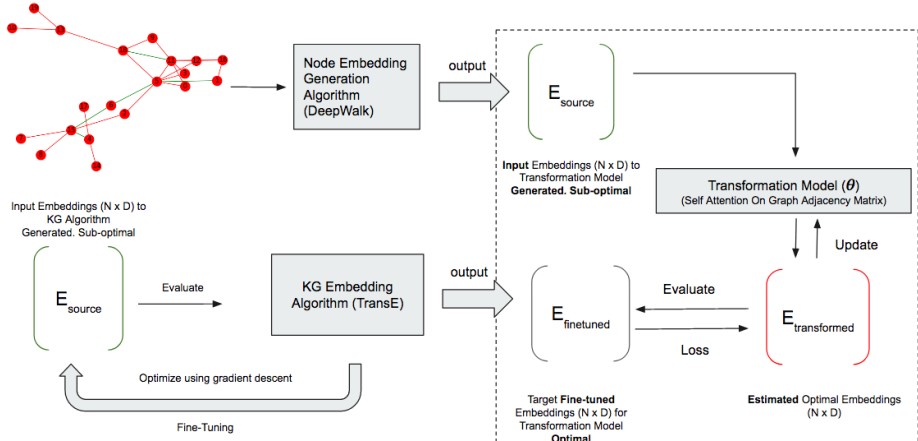

Fig. 1: Transformation Model. Input Graph: Green edges are missing links and red edges represents present links. First, a random walk method outputs node embeddings (source) for a graph. These embeddings are then used to initialize KG embedding method, which outputs finetuned embeddings. A transformation model is then trained between source and finetuned embeddings.

## 3   Methodology

Transformation model is suggested to expedite fine-tuning process with KG-embedding methods. Let $G_{n,m}$ be a graph with $n$ vertices and $m$ edges. Given the node embeddings of the graph $G$, we would want to transform them to optimal node embeddings.

### 3.1   Node Embedding Generation

The input graph $G_{n,m}$ is fed into one of the random walk based graph embeddings methods (node2vec [13] or DeepWalk [25]), which gives us the node embeddings. Let $f$ be a random walk based graph embedding method and $E^i_{source}$ denotes the output node embeddings:

$$E^i_{source} = f(G^i) \tag{1}$$

where $G^i$ is the $i^{th}$ graph in the dataset of graphs $D = \{G^1, G^2, ...\}$ and $E^i_{source} \in \mathbb{R}^{n \times d}$ with the embedding dimension $d$.

### 3.2   Knowledge Embedding Generation

In a KG-based embedding algorithm (such as TransE), the input is a graph and the initial embeddings are randomly initialized. The algorithm uses a scoring function and optimizes the initial embeddings to output the trained embeddings for the given graph. Since we are working with homogeneous graph with only

one type of relation, we don't need to learn the embeddings for the relation, hence they are kept constant and only node embeddings are learnt. Let $E^i_{initial}$ be the initial node embeddings, $E^i_{target}$ be the trained embeddings and $g$ the KG method with parameters $\alpha$.

$$E^i_{target} = g(G^i, E^i_{initial}; \alpha) \tag{2}$$

where $E^i_{target} \in R^{n \times d}$ and $E^i_{initial} \in R^{n \times d}$.

Instead of using randomly initialized embeddings $E^i_{initial}$ to obtain target embeddings $E^i_{target}$, we can initialize with $E^i_{source}$ in Eq. (1) as

$$E^i_{finetuned} = g(G^i, E^i_{source}; \alpha) \tag{3}$$

where $E^i_{finetuned} \in R^{n \times d}$ are fine tuned output embeddings. This idea of better initialization has also been explored previously in [22,10] where it has been shown to result in embeddings of higher quality.

### 3.3 Transformation Model with Self-Attention

Using the node embeddings $E^i_{source}$ from Eq. (1) and fine-tuned KG embeddings $E^i_{finetuned}$ from Eq. (3), we train a transformation model which can learn to transform the node embeddings from a node-based method to KG embeddings. We adopt self-attention [29] on graph adjacency matrix as explained in Algorithm 1:

$$E^i_{transformed} = SelfAttention(G^i, E^i_{source}; \theta) \tag{4}$$

where $E^i_{transformed} \in R^{n \times d}$ are the transformed embeddings and $\theta$ are the parameters of the self-attention model.

The error between the fine-tuned and transformed embeddings is calculated using squared euclidean distance as:

$$E^i_{error} = 1/n \sum ||E^i_{transformed} - E^i_{finetuned}||^2. \tag{5}$$

The loss on batch $\mathbf{X}$ of graphs is measured as:

$$Loss(\mathbf{X}) = 1/b \sum_{i=1}^{b} E^i_{error} \tag{6}$$

where $\mathbf{X} = \{(E^i_{transformed}, E^i_{finetuned})\}$ and $b$ is the batch size. Since KG embeddings are trained from facts/triplets which are obtained from the adjacency matrix of the graph, a self-attention model reinforced with information of the adjacency matrix when applied to node-embeddings is able to learn the transformation function as observed in our experiments (Figure 3). The proposed algorithm is summarized in Algorithm 2.

---

**Algorithm 1:** Self-attention on graph adjacency matrix

---

**1 Function** $SelfAttention(G_{n,m}, E_{n \times d})$
**2**     $A_{n \times n}$ = Adjacency Matrix of $G_{n,m}$
**3**     $K_{n \times d}$ = affine(E, d)
**4**     $Q_{n \times d}$ = affine(E, d)
**5**     $Logits_{n \times n}$ = matmul(Q, transpose(K))
**6**     $AttendedLogits_{n \times n}$ = Logits + A
**7**     $V_{n \times d}$ = affine(E, d)
**8**     $Output_{n \times d}$ = matmul(AttendedLogits, V)
**9**     **return** Output

---

**Algorithm 2:** Training the transformation model

---

**Input:** Dataset of Graphs $D_{train} = \{G^1, G^2, ..., G^n\}$

**1 foreach** $G^i$ *in* $D_{train}$ **do**
**2**     $E^i_{source} \leftarrow f(G^i)$
**3 end**
**4 foreach** $G^i$ *in* $D_{train}$ **do**
**5**     $E^i_{finetuned} \leftarrow g(G^i, E^i_{source}; \alpha)$
**6 end**
**7 while** *true* **do**
**8**     $\mathbf{B} = \{(E^i_{source}, E^i_{finetuned})\}$                  ▷Sample batch
**9**     **foreach** $E^i_{source}$ *in* $\mathbf{B}$ **do**
**10**       $E^i_{transformed} = SelfAttention(G^i, E^i_{source}; \theta)$
**11**     **end**
**12**     $\mathbf{X} = \{(E^i_{transformed}, E^i_{finetuned})\}$
**13**     $\theta \leftarrow \theta - \beta \nabla_\theta Loss(\mathbf{X})$                      ▷Update
**14 end**

---

## 4 Experiments

### 4.1 Datasets

Yang, et. al [33] introduced social network datasets with ground-truth communities. Each dataset $D$ is a network having a total of $N$ nodes, $E$ edges and a set of communities (Table 1).

| Dataset | Description | Nodes | Edges | Communities |
|---|---|---|---|---|
| YouTube | Friendship | 1,134,890 | 2,987,624 | 8,385 |
| DBLP | Co-authorship | 317,080 | 1,049,866 | 13,477 |
| Amazon | Co-purchasing | 334,863 | 925,872 | 75,149 |
| LiveJournal | Friendship | 3,997,962 | 34,681,189 | 287,512 |
| Orkut | Friendship | 3,072,441 | 117,185,083 | 6,288,363 |

Table 1: Datasets

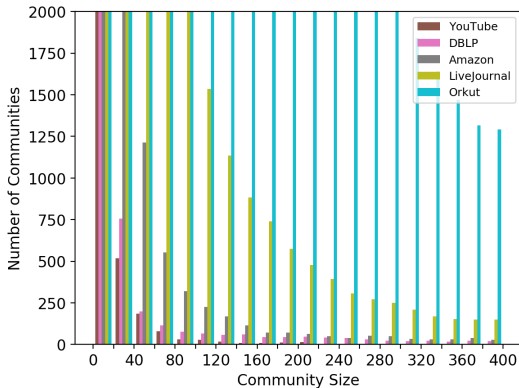

Fig. 2: Histogram showing community size vs its frequency. DBLP, YouTube and Amazon datasets have smaller size communities and LiveJournal and Orkut have larger size communities.

The communities in each dataset are of different sizes. They range from a small size (1-20) to bigger sizes (380-400). There are more communities with small sizes and their frequency decreases as their size increases. This trend is depicted in Figure 2.

YouTube[3], Orkut[3] and LiveJournal[3] are friendship networks where each community is a user-defined group. Nodes in the community represent users, and edges represent their friendship.

DBLP[3] is a co-authorship network where two authors are connected if they publish at least one paper together. A community is represented by a publication venue, e.g., journal or conference. Authors who published to a certain journal or conference form a community.

Amazon[3] co-purchasing network is based on *Customers Who Bought This Item Also Bought* feature of the Amazon website. If a product $i$ is frequently co-purchased with product $j$, the graph contains an undirected edge from $i$ to $j$. Each connected component in a product category defined by Amazon acts as a community where nodes represent products in the same category and edges indicate that we were purchased together.

### 4.2   Training

We consider each community in a dataset as an individual graph $G_{n,m}$ with vertices representing the entity in the community and edges representing the relationship. For training the transformation model, we select communities of particular size range which acts as dataset $D$ of graphs (Table 2). We randomly disable 20% of the links (edges) in each graph to act as missing links for link prediction. In all the experiments, the embedding dimension is set to 32, which

---

[3] http://snap.stanford.edu/data/index.html#communities

| Dataset | Graph Size | Number of Graphs | Average Degree | Average Density |
|---|---|---|---|---|
| YouTube | 16-21 | 338 | 3.00 | 0.17 |
| DBLP | 16-21 | 654 | 4.93 | 0.29 |
| Amazon | 21-25 | 1425 | 4.00 | 0.18 |
| LiveJournal | 51-55 | 1504 | 6.11 | 0.12 |
| LiveJournal | 61-65 | 1101 | 7.20 | 0.11 |
| LiveJournal | 71-75 | 806 | 7.53 | 0.10 |
| LiveJournal | 81-85 | 672 | 6.58 | 0.08 |
| LiveJournal | 91-95 | 497 | 8.01 | 0.08 |
| LiveJournal | 101-105 | 400 | 6.85 | 0.06 |
| LiveJournal | 111-115 | 351 | 5.89 | 0.05 |
| LiveJournal | 121-125 | 332 | 7.67 | 0.06 |
| Orkut | 151-155 | 1868 | 7.20 | 0.04 |
| Orkut | 251-255 | 654 | 7.21 | 0.028 |
| Orkut | 351-355 | 335 | 7.33 | 0.020 |

Table 2: Selected datasets and graph size for experiments.

works best in our pilot test. We used OpenNE[4] for generating node2vec and DeepWalk embeddings and OpenKE [14] for generating KG embeddings. The dataset $D$ of graphs is split into train, validation and test split of 64%, 16%, and 20% respectively.

### 4.3 Evaluation Metrics

For evaluation, we use MRR and Precision@K. The algorithm predicts a list of ranked candidates for the incoming query. To remove pre-existing triples in the knowledge graph, filtering operation cleans them up from the list. MRR computes the mean of the reciprocal rank of the correct candidate in the list, and Precision@K evaluates the rate of correct candidates appearing in the top K candidates predicted. Due to space constraints, we only present the results for MRR. Results of Precision@K can be found at our GitHub[5].

## 5    Results & Discussions

From the results depicted in Figure 3, we observe that the target KG embeddings (TransE, TransH, etc.) almost always outperforms random-walk based source embeddings (node2vec and DeepWalk) except in case of SimplE and DistMult where both the methods perform poorly. This can also be observed in Figure 4.

Finetuned KG embeddings achieved better or equivalent performance as compared to target KG embeddings. This can be confirmed by ANOVA test in Figure 4 where there is no significant difference between the MRRs obtained

---

[4] https://github.com/thunlp/OpenNE
[5] https://github.com/ArchitParnami/GraphProject

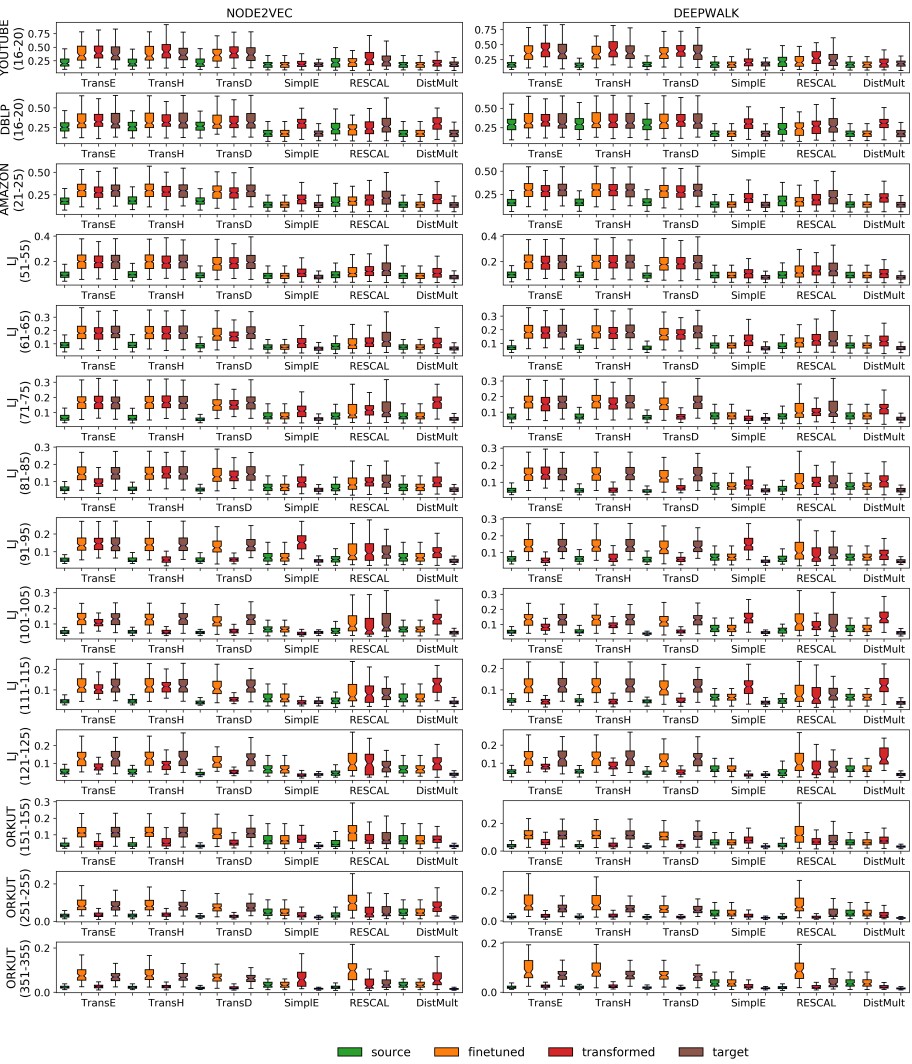

Fig. 3: Performance evaluation of different embeddings on link prediction using MRR (y-axis). Source (green) refers to embeddings from node2vec (left) and DeepWalk (right). Target (brown) refers to KG embeddings from TransE, TransH, TransD, SimplE, RESCAL, or DistMult. For each source and target pair, we evaluate finetuned (orange) embeddings (obtained by initializing target method with source embeddings) and transformed (red) embeddings (obtained by applying transformation model on source embeddings). Results are presented on different datasets of varying graph sizes.

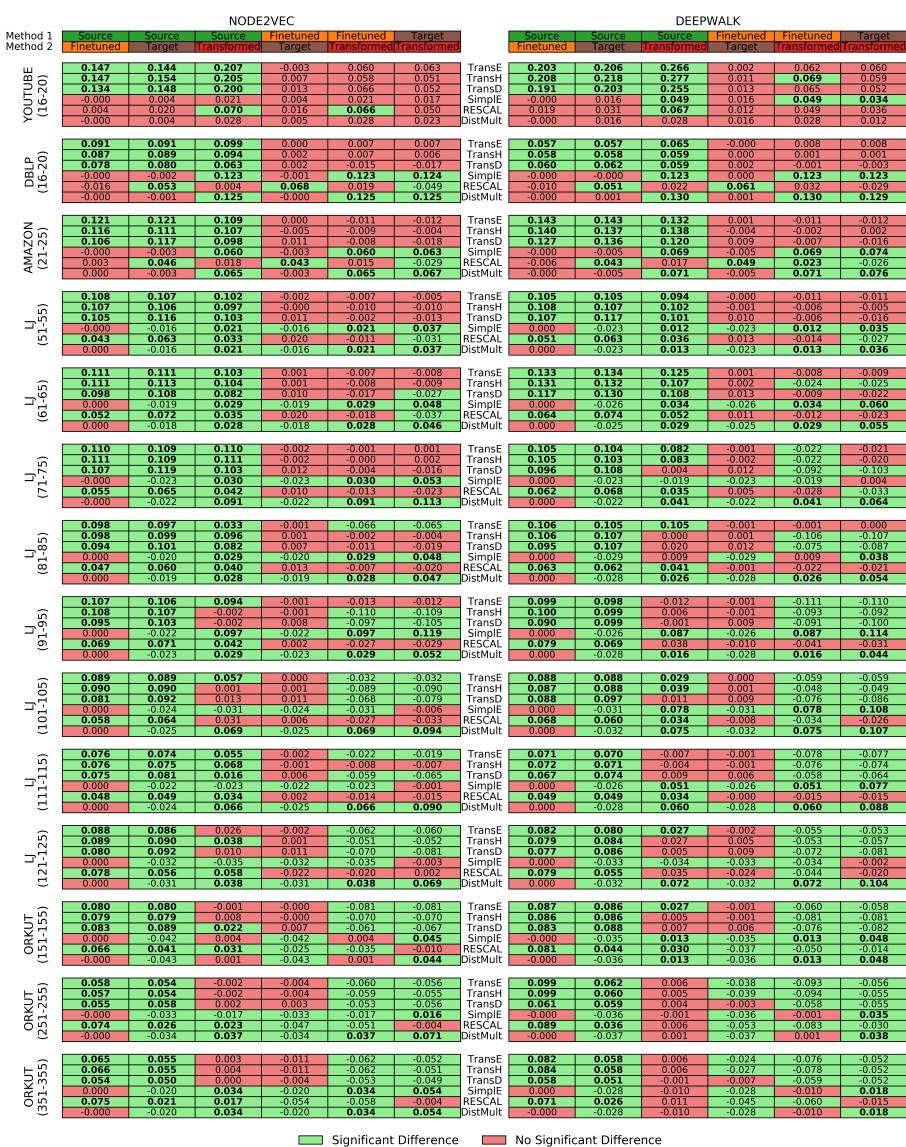

Fig. 4: ANOVA test of MRR scores from two embedding methods (Method 1 and Method 2). The difference of MRR scores between the two methods is significant when their p-values are $<0.05$ (light green) and not significant otherwise (light red). The values in each cell are the difference between the means of MRR scores from two methods (Method 2 − Method 1). The text in bold represents when Method 2 did better than Method 1. Source method refers to node2vec (left) and DeepWalk (right). Target method refers to TransE, TransH, TransD, SimplE, RESCAL, or DistMult in each row.

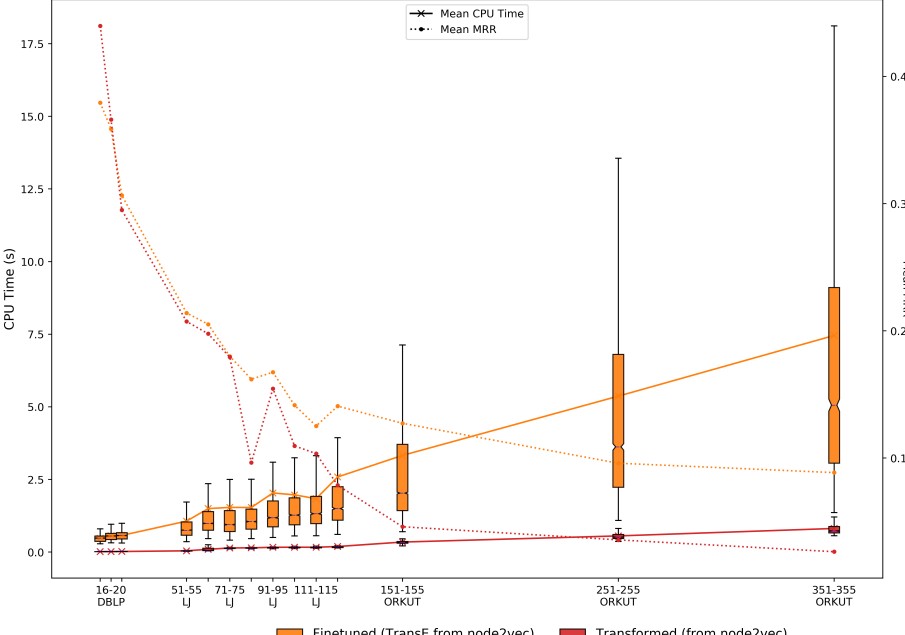

Fig. 5: *CPU Time (left y-axis) vs Graph Size (x-axis)* and *Mean MRR (right y-axis) vs Graph Size* comparison of finetuned (TransE finetuned from node2vec) and transformed embeddings (from node2vec). As the graph size increases the time to obtain embeddings from KG methods (TransE) also increases significantly. However, there is no significant increase in time for the transformation (from node2vec) once we have the transformation model. The Mean MRR scores of both finetuned and transformed embeddings also drop with the increase in graph size, however, they perform equally good (for graphs <76). Note that finetuning time and transformation time both include time to obtain node2vec embeddings as well.

from finetuned and target KG embeddings in most cases. Specifically, translational based methods such as TransE, TransH, and TransD have equivalent performance for finetuned and target embeddings whereas SimplE, RESCAL, and DistMult have better finetuned embeddings than target embeddings as the graph size grows.

Transformed embeddings consistently outperform source embeddings and have similar performance to finetuned embeddings at least for graphs of sizes up to 65. The performance drop starts from graph size 71-75 in the transformation to TransD from DeepWalk whereas 81-85 in the transformation to TransE from node2vec. For RESCAL, the transformation works for larger sized graphs in node2vec and till 121-125 in DeepWalk.

As the graph size increases (top to bottom), the overall MRR scores decrease for all the embeddings as expected. In Figure 5, we compare computation time and MRR performance of transformed embeddings and finetuned embeddings where source method is node2vec and target method is TransE. It can be seen that the transformed embeddings give similar performance as finetuned embeddings (without any significant increase in computational cost) up to graphs of size 71-75. Thereafter the transformed embeddings perform poorly, we attribute this to poor finetuned embeddings on which the transformation model was trained.

## 6    Conclusion

In this work, we have demonstrated that random-walk based node embedding (source) methods are computationally efficient but give sub-optmial results on link prediction in social networks whereas KG based embedding (target & finetuned) methods perform better but are computationally expensive. For our requirement of generating optimal embeddings quickly for real-time link prediction we proposed a self-attention based transformation model to convert walk-based embeddings to optimal KG embeddings. The proposed model works well for smaller graphs but as the complexity of the graph increases, the transformation performance decreases. For future work, our goal is to explore better transformation models for bigger graphs.

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
