# OpenReview forum: "Transformation of Node to Knowledge Graph Embeddings for Faster Link Prediction in Social Networks"
_kg-construct.github.io/KGCW/2022/Workshop — KGCW 2022_

### Official Review · ~Elvira_Amador-Domínguez1 · 2022-04-01
**The proposal is interesting, but relevant technical aspects are left unexplained**

**Rating:** 5
**Confidence:** 4

**Review:**

This paper presents a transformation model to convert random walk obtained node embeddings into knowledge graph embeddings. The idea presented in this paper is interesting and has potential, but there are some crucial flaws in the content and presentation.

Regarding presentation, the paper is well-written and structured, and the English is good. However, Fig. 1 is hardly readable due to the font size. The same applies to Fig 4., where the values in the tables are hardly readable.

My main concern is the lack insight on key aspects. Specifically, regarding scalability, as this is one of the main claims of the proposal. Is it necessary to train three models (random-walk, knowledge graph embeddings and the transformation) or just the transformation? This is a key element to the proposal and should therefore be clearly outlined.

---

### Official Review · ~Femke_Ongenae1 · 2022-04-05
**The proposed method is interesting, but the choices for Knowlege Graph embedding methods and evaluation should be better motivated (or updated) to be in line with the state of the art for link prediction.**

**Rating:** 6
**Confidence:** 4

**Review:**

- The Introduction makes the following claim: "Neither of these two approaches, however, can generate “optimal” embeddings “quickly” for real-time link prediction on new graphs. Random walk based node embedding methods are computationally efficient but give poor results whereas KG-based methods produce optimal results but are computationally expensive."
This claim is made without any references or elaboration to support it. As it is the main problem on which the paper is built, I would expect the authors to better back up this claim.
Other papers already have performed thorough comparisons of the performance of walk-based embedding techniques (node2vec, deepwalk, rdf2vec) with more translational distance models (TransE, TransH, etc.), for example, one of the most recent papers being "Jan Portisch, Nicolas Heist, Heiko Paulheim: Knowledge Graph Embedding for Data Mining vs. Knowledge Graph Embedding for Link Prediction - Two Sides of the same Coin?. Accepted for publication in Semantic Web Journal", which also cites several other works performing comparisons.

- Also in the abstract, it would have been better if these requirements needing to deal with continuously changing graphs and delivering results real-time would have been emphasized more to highlight the need for the research in the paper.

- Why were these specific KG embedding methods chosen for this paper (e.g. TransE, TransH, TransD, , Rescal and SimplE)? Was there any specific motivation for this selection of methods? Why was for example Complex, Tucker, and RotatE not considered which have been showing impressive results on a number of link prediction tasks? In recent papers these techniques seem to achieve the state-of-the-art results on link prediction tasks, so they would seem to be a more motivated choice: Andrea Rossi, Denilson Barbosa, Donatella Firmani, Antonio Matinata, and Paolo Merialdo. 2021. Knowledge Graph Embedding for Link Prediction: A Comparative Analysis. ACM Trans. Knowl. Discov. Data 15, 2, Article 14 (April 2021), 49 pages. DOI:https://doi.org/10.1145/3424672

- Deep learning based models, such as ConvE, were also not considered, why not? Would it be possible to replace the role of the translational model (e.g. TransE) within your technique with a deep learning model? Would this have any benefits? Did you try this?

- Finally, did u consider techniques such as RDF2Vec in your comparison as they already seem to fullfill some of your wishes, i.e. they are computationally less expensive as they are walk-based, but they they take into account all the Knowledge from the Knowledge Graph, and (as shown by the above cited paper amongst others), they also have shown more competitive results on link prediction tasks. If the used walk-based techniques (Node2Vec, deepWalk) would be replaced by RDF2Vec (which also takes into account the Knowledge Graph) in your technique, what would be the impact? Do you think there would be any benefit?

- However, I do believe the method the authors introduce is novel and interesting.

- Overall, I believe this work could merit a presentation at a workshop, however, the paper could use some more framing of the work introducing it as a first preliminary presentation and evaluation of the introduced algorithm. As highlighted above, some references to related work already performing detailed evaluations and comparisons of node / Knowledge graph embeddings for link prediction should be referenced and discussed in the introduction of the paper. Moreover, it should be discussed why some methods were selected, and if a proper motivation cannot be given, maybe the comparison should be updated to also include other methods, such as Complex, Rotate and Tucker.

- I could not find the results at https://github.com/ArchitParnami/GraphProject ?
- Is the code of the algorithm also put open source? I would highly recommend to do so.

Language suggestions:
- Abstract: Recent advances in neural networks have solved common graph problems, such as link prediction, node classification, node clustering, AND node recommendation by developing embeddings of entities and relations into vector spaces. --> insert comma before such as, add AND
- General comment: Add commas before such as
- Introduction: ...where a node represents people or entity... --> ...where a node represents A PERSON or AN entity...
- Introduction: ..., AND Katz Index [16]... (add and)
- Introduction: ...simplify the manipulation, while preserving the inherent structure... (add comma)
- Problem definition: ..., used in [9,6,8] --> please use author et al. when using references as part of a sentence
- Problem definition: In this task, for each position of missing entity, ... --> for each position where an entity is missing
- Problem definition: ...where both h and t are of same kind --> are of the same kind
- Graph embedding methods: title: Node Embeddings using Random Walk --> WalkS (add S)
- Graph embedding methods: ...as detailed in [30]. --> please use author et al. when using references as part of a sentence
- Caption Fig. 1: These embeddings are then used to initialize A KG embedding method, which outputs finetuned embeddings. (add A)
- Methodology: A transformation model is suggested to THE expedite fine-tuning process with KG-embedding methods. (add A and the)
- Please use in text the same label to refer to the figures as used in the caption, i.e. use Fig. instead of Figure.

---

### Official Review · ~Mauro_Dragoni1 · 2022-04-05
**The manuscript is and interesting work that should be presented at the workshop.**

**Rating:** 7
**Confidence:** 4

**Review:**

The manuscript investigates a transformation model which converts node embeddings obtained from random walk based methods to embeddings obtained from knowledge graph methods directly without an increase in the computational cost.
The topic is timely and well motivated.
The paper is well written and quite easy to follow even for people that are not expert in the field since it provides all basic elements for understanding the message the authors want to provide.
I do not have any issue about this work.
The approach is clear motivated and well presented in the paper.
The evaluation is the strong point of this work.
The authors provided a reproducible experimental settings by using publicly available datasets and the methodology they adopted.
The evaluation is very extensive and the results are promising.
I warmly suggest to accept this work for the workshop.

---

### Decision · Program_Chairs · 2022-04-11

**Decision:**

Accept

**Comment:**

Dear authors,

Thank your for submitting your paper. We are happy to inform you that we accept your paper! Please carefully consider the reviews when you prepare your paper for the camera-ready version. You will receive specific instructions to submit your camera-ready soon.

Kind regards
Organizers of the Knowledge Graph Construction workshop 2022